# Evolution of the Electronic and Optical Properties of Meta-Stable Allotropic Forms of 2D Tellurium for Increasing Number of Layers

**DOI:** 10.3390/nano12142503

**Published:** 2022-07-21

**Authors:** Simone Grillo, Olivia Pulci, Ivan Marri

**Affiliations:** 1Department of Physics, University of Rome Tor Vergata and INFN, Via della Ricerca Scientifica 1, 00133 Roma, Italy; 2Department of Sciences and Methods for Engineering, University of Modena e Reggio Emilia, 42122 Reggio Emilia, Italy; 3Interdepartmental Center for Research and Services in the Field of Hydrogen Production, Storage and Use H2—MO.RE, Via Università 4, 41121 Modena, Italy

**Keywords:** Tellurium, Tellurene, Density Functional Theory, ab initio, exciton, chain

## Abstract

In this work, ab initio Density Functional Theory calculations are performed to investigate the evolution of the electronic and optical properties of 2D Tellurium—called Tellurene—for three different allotropic forms (α-, β- and γ-phase), as a function of the number of layers. We estimate the exciton binding energies and radii of the studied systems, using a 2D analytical model. Our results point out that these quantities are strongly dependent on the allotropic form, as well as on the number of layers. Remarkably, we show that the adopted method is suitable for reliably predicting, also in the case of Tellurene, the exciton binding energy, without the need of computationally demanding calculations, possibly suggesting interesting insights into the features of the system. Finally, we inspect the nature of the mechanisms ruling the interaction of neighbouring Tellurium atoms helical chains (characteristic of the bulk and α-phase crystal structures). We show that the interaction between helical chains is strong and cannot be explained by solely considering the van der Waals interaction.

## 1. Introduction

Since the advent of graphene [1], much effort has been devoted to the search of two-dimensional (2D) layered materials, which can often be obtained from layered van der Waals (vdW) solids. Due to the naturally terminated surface with vdW interactions—rather than dangling bonds—these 2D materials are generally stable in ambient conditions, offering extraordinary mechanical, electrical and optical properties. Thanks to the quantum confinement effect, 2D materials possess distinct characteristics from their corresponding bulk counterparts, thus receiving wide attention from science and industry. The possibility of further modifications by means such as stacking, doping, twisting, gating, etc., provides new potential in the application to electronics, optoelectronics and energy storage devices, superconductors and so on. Up to now, a large number of 2D materials has been discovered and fabricated. In particular, elemental 2D materials have gained special interest in the last years. So far, at least 15 types of elemental 2D materials have been experimentally realised or theoretically predicted [2,3,4,5,6,7,8,9,10,11,12,13,14].

The subject of this work, Tellurene, is a new-emerging elemental 2D material, with fascinating electronic and optical properties, dramatically differing from its bulk counterpart, which has come to us owing to its unique chained structure. Tellurene is the 2D form of Tellurium (group VI-A), whose existence was first predicted in 2017 by Zhu et al. [14] and then verified by several experimental analysis [15,16,17].

Tellurene (Te) simultaneously overcomes shortcomings such as the zero-bandgap of graphene, the air instability of black phosphorus and the small carrier mobility of MoS${}_{2}$. In further explorations, it was found that Te or Te-based devices present excellent thermoelectric properties, piezoelectric properties, quantum Hall effect, high carrier mobility and superb optical properties—especially nonlinear optics characteristics—and many others. Similarly to other 2D materials, electronic and optical properties of Te can be modulated by virtue of strain, defects, edges, substrate-induced modulations and so on [18]. All these interesting features, along with a proven good environmental stability, are critical for exploring the fundamental properties and the technological prospects of Te or Te-based devices [19], such as field effect transistors (FETs) [20,21] and chemical sensors [22,23,24] above all, as well as many others [25,26,27,28,29].

Monolayer (ML) Te can exist in three different allotropic forms, i.e., the most stable 1T-MoS${}_{2}$-like structure (γ-Te), the metastable orthorhombic (β-Te) and 2H-MoS${}_{2}$-like (ε-Te) structures [14]. Noticeably, ML α-phase—which is characterised by parallel helical chains, similarly to the stable bulk Te-I phase—is unstable and can be transformed into ML β-phase without barrier [30]. From a practical point of view, α-, β- and γ-phase exhibit very interesting physical properties, good environmental stability and can be more importantly feasibly fabricated using different experimental techniques [15,17,31,32]. For these reasons, in this work we focus our attention on these three allotropic forms of 2D Te. Here, the different phases will be identified following the notation introduced in Refs. [33,34], where the alphabetical order represents the formation energy of these phases, above the ML, in ascending order, in contrast with the notation of Ref. [14], where α-Te and γ-Te stand for our γ-Te and ε-Te, respectively.

In addition to MLs mentioned above, further theoretical calculations have shown that α-Te should be the most stable phase if the thickness is beyond 1 layer [30], leading directly to the formation of bulk Te-I for increasing number of layers.

The main intent of this work is to provide a better understanding of the evolution of the physical properties of Te for increasing number of layers. By means of ab initio calculations, using Density Functional Theory (DFT), structural relaxations, electronic bandstructures and optical absorption calculations have been carried out systematically for three different allotropic forms (α-, β- and γ-phase), for increasing number of layers (Figure 1). Our studies confirm the results of Refs. [14,30] and provide new outcomes concerning aspects such as stacking, stability and geometry configurations of 2D Te, the dependence of electronic bandstructures and optical absorption spectra on the number of layers (with newfound values of the gaps for the α-phase) and, using an analytical model for 2D materials [35,36], we evaluate the excitonic binding energies and radii [37,38,39] of the studied systems. Finally, in the case of the bulk and α-phase, novel details concerning the interaction between helical chains of Te atoms are provided.

## 2. Materials and Methods

ab initio calculations have been performed, within the DFT framework, using the GGA-PBE exchange-correlation functional, as implemented in the Quantum ESPRESSO (QE) integrated suite [40,41]. Norm-conserving, full-relativistic pseudopotentials with a kinetic cutoff of 105 Ry have been adopted for all the considered structures. All calculations have been performed with and without the inclusion of spin-orbit corrections (SOC). For every system, a full-structure relaxation has been carried out, including different van der Waals (vdW) corrections, in order to provide optimised lattice parameters. Through a comparison with previous works in the literature, the Grimme’s DFT-D2 vdW correction [42] was found to be the most suitable. In order to investigate—properly ab initio—the nature of the of the interaction between the Te atoms helical chains (characteristic of the bulk and α-phase crystal structures), the Tkatchenko-Scheffler vdW dispersion correction [43] has also been used. Electronic bandstructures and optical absorption spectra have been calculated at the single particle level. Convergence tests were separately conducted on both the k-points mesh in the Brillouin Zone (using Monkhorst-Pack grids) and empty electronic bands (for the optical spectra). A vacuum thickness of at least 15 Å has been considered to separate each replica in the out-of-plane direction.

## 3. Results

### 3.1. Geometry and Stability

As mentioned above, Zhu et al. [14] predicted three (meta-)stable phases for 2D ML Te, namely, β-, γ- and ε-phase, while ML α-phase is unstable and can be transformed into β-phase without barrier. In this work, the attention is focused on the more stable and interesting β- and γ-phase — concerning the ML form — also including α-phase for increasing number of layers, up to a number of 4. In this sense, we provide a complete and systematic analysis of the evolution of the total energy per atom of the systems (as shown in Figure 2), also reporting new quantitative results regarding stability, stacking configurations and lattice parameters.

Starting from the results obtained for a single layer, our structural relaxations confirmed that α-phase is unstable and spontaneously evolves towards the β-phase structure. The optimised lattice parameters of ML β- and γ-Te are shown in Table 1. The obtained results are in very good agreement with other theoretical outcomes [14,44]: remarkably, in the case of β-Te, starting from an orthorhombic conventional cell with 3 Te atoms, the axes are slightly tilted, losing all its internal symmetries; γ-Te, on the contrary, possesses a conventional cell with hexagonal symmetry, containing 3 Te atoms. The γ-phase appears to be the most stable, with an energy gain of about 53 meV per atom.

We discuss now the results obtained for the Te bilayer (2L) structures. In this work, two different configurations have been studied for both β- and γ-phase, corresponding to the AA and the AB stacking. For what concerns β-Te, the AB stacking is found to be more stable than the AA pattern, with an energy difference of about 16 meV per atom. On the contrary, for γ-Te, the AA stacking results to be the most stable configuration, with an energy difference of about 30 meV per atom with respect to the AB arrangement. The 2L γ-phase still maintains its hexagonal symmetry structure, with 6 atoms per unit cell. Overall, 2L β-phase is more stable than the γ-phase, with an energy difference of 36 meV per atom. In addition, in this case, the optimised lattice parameters, reported in Table 1, are in fairly good agreement with those reported in Ref. [45]. Interestingly, the presence of two layers makes the formation of the α-phase energetically sustainable, which, in this configuration, results to be more stable than both β- and γ-phase, with an energy difference with β-phase of about 10 meV per atom (see Figure 2). The conventional cell is orthorhombic, with 6 atoms forming two layers of shifted parallel helical chains.

Starting from 2L α-Te and following an alternate pattern of parallel helical chains, one can construct three-layer (3L) and four-layer (4L) α-Te: indeed, for increasing number of layers, the stability of the α-phase becomes prevalent, eventually leading to bulk Te-I: the energy difference with the β-phase increases to 13 meV per atom (3L) and 16 meV (4L). The optimised lattice parameters, reported in Table 1, are in very good agreement with the results of Qiao et al. [30], possibly the only reference concerning ground-state properties of few-layer (FL) α-Te (obtained with optB86-vdW functional). The 3L and 4L lattice parameters of β- and γ-phase are also reported in Table 1.

### 3.2. Electronic Bandstructures

The electronic bandstructures of all three phases have been calculated for an increasing number of layers, with and without the inclusion of SOC. We have found out that SOC are essential for a proper description of the electronic properties of Te; hence, here we present only bandstructures with SOC included. The evolution of the electronic bandgap for increasing number of layers is displayed in Table 2.

In the case of the β-phase, bandstructure calculations were performed along the high-symmetry directions given by the k-path Y→Γ→X→S→Y in the orthorhombic 2D Brillouin Zone (BZ) (Figure 3 and Figure 4). Interestingly, in ML β-Te (as well as in 2L and 3L) the inclusion of SOC induces a transformation from an indirect to a direct bandgap at the Γ point, in agreement with Zhu et al. [14]. By increasing the number of layers, the electronic bandgap of the β-phase decreases from 1.02 eV (in agreement with Refs. [14,46]), in the ML case, to 0.31 eV (2L), 51 meV (3L) and 6 meV (4L), in which the valence band maximum (VBM) and the conduction band minimum (CBM) move away from the Γ point, giving rise to an indirect bandgap.

Regarding the γ-phase, the calculations were performed along the high-symmetry directions Γ→M→K→Γ in the hexagonal 2D BZ (Figure 3 and Figure 5). In this case, the inclusion of SOC does not modify the nature of the gap, which remains indirect, but still moves the position of the VBM from the K→Γ to the Γ→M direction, with the CBM remaining fixed at Γ, except for 4L γ-Te, which apparently undergoes a semiconductor-to-metal transition, with the conduction band exceeding the valence band at Γ. In addition, also in this case, the electronic bandgap decreases with increasing number of layers, going from 0.42 eV (ML, in agreement with literature [14]) down to 0.15 eV (2L), 26 meV (3L) and then vanishing with a semiconductor-to-metal transition when the number of layers is equal to 4 (see Table 2). This metallic transition may be due to the fact that the γ-phase is expected to completely transforms—at 5 layers—into a further phase, called $\alpha^{\prime }$ (which however is still less stable than β- and α-phase) [30] and it is thus subject to increasing strain.

In Figure 6, we report the bandstructures calculated for the α-phase, considering a standard high-symmetry directions path Γ→X→S→Γ in the orthorhombic 2D BZ. Starting from the bandstructures of Figure 6, we extrapolate GGA-PBE indirect energy gaps of 0.78 eV (2L), 0.64 eV (3L) and 0.46 eV (4L), which are similar to the ones of Ref. [30], obtained by using the same functional, that is, 0.71, 0.52 and 0.44 eV, respectively. However, this path is not the best choice to evaluate the bandgaps of FLs α-Te. Indeed, a very dense sampling of the BZ has revealed that both the VBM and CBM should be found out of high-symmetry directions—approximately around k-point (0.21, 0.20, 0), nearby the X→Y direction—for all the layers, giving overall novel lower values of the electronic bandgaps, that is, 0.67 eV (2L), 0.50 eV (3L) and 0.42 eV (4L). As in the previous cases (see Table 2 and Figure 3), the electronic bandgap decreases for increasing number of layers, slowly tending to the estimated PBE-SOC limit of bulk Te-I of about 30 meV [46]. Band-splitting, due to SOC, lowers the value of the electronic bandgaps without changing their nature, which remain indirect. Overall, the inclusion of SOC makes α-phase bandstructures very dense and tangled, with the emergence of recurring linear or nearly-linear band dispersions below and above the Fermi level, especially at high-symmetry Γ and *X* points.

### 3.3. Optical Absorption

We now move on to discuss the optical properties (in particular, the optical absorption) of the three studied phases for increasing number of layers, with SOC included. A detailed convergence of the optical spectra requires the use of dense k-mesh grids, that is: 180×180×1 for MLs, 120×120×1 for 2Ls, 90×90×1 for 3Ls and 45×45×1 for 4Ls. Due to their unique features, results obtained for the ML β- and γ-Te are discussed separately.

From the linear response theory for a homogeneous medium, the absorption coefficient A(ω) is given by:(1)$$A\left(\omega \right)=\frac{\omega }{c}\varepsilon_{2}\left(\omega \right)L_{z}$$
where *c* is the speed of light in vacuum, $\varepsilon_{2}\left(\omega \right)$ is the imaginary part of the dielectric function, defined as ${\tilde{n}}^{2}\left(\omega \right)=\varepsilon \left(\omega \right)=\varepsilon_{1}\left(\omega \right)+i\varepsilon_{2}\left(\omega \right)$, and $L_{z}$ is the length of the supercell in the (out-of-plane) z-direction. In this way, the calculated absorbance is independent of the size of the vacuum in the z-direction of the periodic supercell. In the dipole approximation, $\varepsilon_{2}\left(\omega \right)$ can be defined—save for a constant factor—through the Fermi’s golden rule as:(2)$$\varepsilon_{2}\left(\omega \right)\propto \frac{1}{V\omega^{2}}\sum_{c,v}\sum_{k}{\left|\psi_{c,k}\hat{e}\cdot p\psi_{v,k}\right|}^{2}\;\delta \left(E_{c,k}-E_{v,k}-\hbar \omega \right).$$

In our approach, absorption spectra are calculated without including phonon contributions, that is, only direct transitions are permitted (i.e., q=0). As a consequence, the absorption energy threshold is uniquely determined by the direct energy gap of the system. Moreover, we neglect quasiparticle and excitonic effects, which will be the subject of a further work.

Here, $\varepsilon_{2}\left(\omega \right)$ is computed using the pw2gw tool of the QE package, which gives separate contributions in the axis directions; then, by taking the mean value of the *x* and *y* components, the in-plane optical absorbance has been calculated starting from Equation (Equation 1).

Optical absorption spectra calculated within the single-particle approximation for the ML β- and γ-Te are reported in Figure 7. Due to the smaller direct energy gap, γ-Te shows a lower absorption energy threshold than β-Te. Moreover, it shows a more intense optical response in the range of energies 0.8–4 eV. In β-Te (Figure 7a), intense peaks stick out roughly between 1.5 and 2 eV and between 2.5 and 3 eV. Interestingly, we found that ML β-Te shows a strong optical anisotropy at low photon energies. In fact, an analysis of the dipole matrix elements (see Equation (Equation 2)) reveals that the peak at around 1.5 eV is mainly due to transitions for light polarised along the in-plane *y* direction, which thus gives a major contribution to the mean overall behaviour.

Increasing the number of layers, the optical response gradually grows, especially in the case of the γ-phase (Figure 8a), which shows a very large absorbance between 1 and 2 eV, with two very intense peaks at about 1 and 1.6 eV (4L) as well as a non-neglibile activity between 7 and 8 eV; as for β-phase (Figure 8b), a significant optical absorption appears to start at very low energies, with a very pronounced peak (for 3L and 4L) at around 1 eV. Very interestingly, β-phase appears to almost completely lose its optical anisotropy when the number of layers is increased, despite its crystal structure. The general reduction of the energy gap with increasing number of layers (see Table 2) leads, for both phases, to a lowering of the absorption energy threshold, which reach the lower value for the metallic 4L γ-Te system (see Table 2).

Finally, the optical absorption of the α-phase is shown in Figure 8c. Also in this case, the absorption energy threshold decreases with increasing number of layers (see Table 2). This effect is associated to a non-negligible increase of the optical response in the range of energy up to 3 eV.

### 3.4. 2D Exciton Model

In this section, we apply analytical methods, introduced by Keldysh [35] and Rytova [36], for estimating, as a good approximation, the exciton binding energies ($E_{B}$) and exciton radii ($r_{ex}$) of 2D-Te. These methods have been already successfully applied to hydrogenated group IV 2D sheets (graphane, silicane, germanane, and hydrogenated SiC) [38] and group III 2D ML nitrides [39].

In the limit of isolated sheets of vanishing thickness, that is, 2D structures formed by few layers within a simulation box, sufficiently large to avoid spurious interaction between replicas, the statically (ω→0) screened electron-hole interaction can be described by the Rytova-Keldysh potential, that is:(3)$$W\left(\rho \right)=-\frac{\pi e^{2}}{2\rho_{0}}\left[H_{0}\left(\frac{\rho }{\rho_{0}}\right)-N_{0}\left(\frac{\rho }{\rho_{0}}\right)\right]$$
where $H_{0}$ is the Struve function, $N_{0}$ is the Neumann-Bessel function of the second kind, and $\rho_{0}$ is the screening radius $\rho_{0}=2\pi \alpha_{2D}$, with $\alpha_{2D}$ the static electronic sheet polarizability, given by:(4)$$\alpha_{2D}=\frac{L}{4\pi }\varepsilon_{2}^{∥}\left(0\right).$$

In Equation (Equation 4), *L* is the cell thickness along the periodic direction (z-axis) and $\varepsilon_{2}^{∥}\left(0\right)$ is the immaginary part of the in-plane zero-frequency dielectric function, that can be calculated using an independent particles approach, as discussed in Section 3.3.

The exciton binding energies and radii can be determined by solving a 2D Schrödinger-like equation:(5)$$\left\{E_{G}-\frac{\hbar^{2}}{2\mu }\nabla_{\rho }^{2}+W\left(\rho \right)\right\}\phi_{n}\left(\rho \right)=E_{n}\phi_{n}\left(\rho \right)$$
which describes the internal motion of an electron-hole pair with a reduced mass μ, at a distance ρ, in a material with energy gap $E_{G}$. This simple, single particle approach of the electron-hole problem generally provides a rather reliable qualitative (and even quantitative) description of the studied systems, regarding the lowest-energy excitons. Equation (Equation 5) can be analytically solved only in the limit $\rho /2\pi \alpha_{2D}\gg 1$ [47,48]. Otherwise, by applying a variational approach with a trial wavefunction, we can obtain an analytical expression for both $E_{B}$(λ) and $r_{ex}$(λ), which depend explicitly on the variational parameter λ and also through the parameter $\beta \left(\lambda \right)=a_{ex}/8\pi \lambda \alpha_{2D}$, where $a_{ex}=a_{B}m/\mu$ is the Bohr radius renormalised by the effective reduced mass μ in the units of the electron mass *m* (see Ref. [38] for more details). The exciton binding energy $E_{B}$(λ) has to be maximised with respect to λ and can then be calculated for different values of $4\pi \alpha_{2D}/a_{ex}$. The result is shown in Figure 9. The binding energy (red solid line) decreases with increasing polarizability $\alpha_{2D}$, while the exciton radius (blue solid line) increases. The left-hand limit (β(λ)≫1) corresponds to having an unscreened 2D Coulomb potential in Equation (Equation 5), i.e., $W\left(\rho \right)=-e^{2}/\rho$. On the other hand (β(λ)≪1), we have a logarithmic behaviour for the screened interaction ($W\left(\rho \right)\propto ln\left(4\pi \alpha_{2D}/\rho \right)$).

For the considered 2D Te structures, the electron (hole) effective mass has been calculated starting from the electronic bandstructure, evaluated nearby the CBM (VBM) (see [49] for more details). The obtained results, averaged in the x and y directions and renormalised by *m*, are reported in Table 3 for all the considered systems, except for metallic 4L γ-Te. Here, it is worth highlighting that, while γ-phase exhibits a perfect isotropy in the two in-plane directions, α- and (especially) β-phase display a noteworthy anisotropy, with higher values of the effective mass in the *x* direction.

The estimated values of the exciton binding energies $E_{B}$ are reported in Table 3. When possible, they are compared with the results obtained within GW-BSE calculations [46,50], showing a very good agreement. Further to this point, we just point out that, in the case of ML β-Te, Pan et al. [46] performed calculations, without SOC, with polarized light, obtaining a binding energy of $E_{B}$ = 0.47 eV in the *x* direction and $E_{B}$ = 0.67 eV in the *y* direction [46], thus an in-plane mean value of about $E_{B}$ = 0.57 eV. To the best of our knowledge, there is no such calculation for FL α-Te.

As expected, by increasing the number of layers, we observe a decreasing of the exciton binding energy. This is due to the related decreasing of the bandgap, accompanied by a more effective electronic screening. Moreover, we generally observe a reduction of the exciton reduced mass for increasing number of layers. This results in an increasing of the exciton radius, as shown in Table 3.

Physically, the model allows us to get insights into the features of the systems: as shown in Figure 9, the values we obtained lie beyond those of other previously studied systems [37,39], towards the logarithmic limit (β(λ)≪1). This means that, in our case, the bound state formed by the electron-hole pair cannot be described in terms of a quasiparticle in a hydrogen-like, 2D unscreened Coulomb potential (β(λ)≫1). We can also deduce that these systems should possess an important electronic screening, thus showing weakly bound, widespread excitons.

### 3.5. Tellurium Interchain Interaction

The most stable structure of bulk Te at room pressure is trigonal and it is known as Te-I. It consists of helical chains parallel to the c-axis, which are arranged in a hexagonal array. This arrangement characterises also the FL α-Te, which shows a structure organised in shifted layers of parallel chains (see Figure 1). In order to properly study the structural properties of these systems, it is necessary to investigate the mechanisms ruling the interaction between these helical chains.

In Te structures, chemical bonds are mainly formed through the involvement of 5p states. For each Te atom in a chain, two p-electrons covalently bond with two adjacent Te atoms along the chain, while lone pairs of electrons (the remaining two p-electrons of each Te) are allocated between the chains. This kind of electronic coupling should give rise to strong interactions along the chain (intrachain interaction) as well as to weaker interactions between neighbouring chains (interchain interactions).

A vdW-like force was initially invoked to describe the interchain interaction between chained Te layers [51]. This picture was however argued by Yi et al. [52] that, starting from the results obtained by measurements of electrical resistivity, Hall coefficient, and thermoelectric power [53], proposed a scheme where the interchain interaction was ascribable to the formation of weak chemical bonds between the electron lone pairs of Te atoms of neighbouring chains. In this picture, each Te atom in the chain behaves as both an electron acceptor and an electron donor to and from the neighbouring chains, creating an interchain bond relatively weaker than the covalent ones along the chains. Nonetheless, the actual nature of this interchain interaction remains debated.

As a starting point in our investigation, we adopted the analysis proposed by Alvarez [54], which establishes robust geometric and bonding criteria to identify elements and compounds characterised by vdW-like interactions. Noticeably, this approach was also adopted by Marzari et al. [55] to predict, by high-throughput calculations, a set of novel, stable and promising 2D materials. By extracting, for a given couple of elements, a vdW peak (see Figure 10) from an histogram sampling the experimental bonding distances, and by introducing the vdW radii sums as the point of maximum slope of the vdW peak (e.g., the distance corresponding to the full width half maximum of the vdW peak), S. Alvarez defines a simple semi-quantitative guide to address a given intermolecular distance between an atom pair, that is:interatomic distances between ±0.7 Å the vdW radii sum fall into the vdW peak, while longer distances should indicate non-interacting atoms;distances shorter than the vdW radii sum by more than about 1.3 Å correspond most likely to a chemical bond, and those between 0.7  to 1.3 Å shorter fall within the so called “vdW gap” (see Figure 10), thus suggesting a special bonding situation that asks for a deeper analysis.

**Figure 10 nanomaterials-12-02503-f010:**
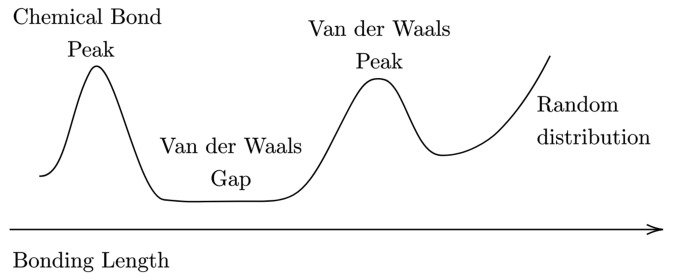
Pictorial scheme of a general atom-atom bonding length distribution, as described by Alvarez [54].

The above analysis is unambiguously valid whenever the chemical bond peak and the vdW peak (both with a proper width) are clearly discernible, that is, when they are separated by a vdW gap, a range of distances at which practically no other peaks are found. This is qualitatively schematised in Figure 10. Otherwise, this scheme cannot be somehow predictive. In our case of interest, Alvarez found a non-neglibile superposition between the tails associated to the above-mentioned peaks; that is, the Te-Te interaction distribution is characterised by a “pseudo” vdW gap. Considering the calculated average interchain equilibrium lengths of 2L, 3L and 4L α-Te, which are, respectively, 3.25, 3.31 and 3.34 Å, they clearly appear to be borderline within the 0.7 spread around 3.98 Å, which corresponds to the Te vdW radii sum given by Ref. [54]. Thus, a simple analysis of the interchain distance, starting from the systems under study, does not unambiguously clarify the role played by the vdW interaction in our situation and a different approach is needed.

In order to deepen our understanding on this matter, we performed DFT calculations for a pair of helical chains, adopting both (i) semi-empirical Grimme’s DFT-D2 [42] and the (ii) ab initio Tkatchenko–Scheffler vdW dispersion correction [43].

Preliminarly, structural properties of a single chain consisting of a unit cell of 3 Te atoms have been converged within the DFT framework, both with and without the inclusion of vdW corrections. Then, in order to investigate the role played by the vdW corrections in the interchain distance, two different simulations have been performed. First, we have followed the most simple approach, in which two parallel chains have been rigidly and gradually moved closer and studied with and without the inclusion of the Grimme’s vdW correction. The results of this analysis is shown in Figure 11, where the interchain binding energy is reported as a function of the interchain distance and the equilibrium interchain separation is highlighted. Secondly, we refine our approach, by letting the whole system to fully relax at each chosen interchain distance, with and without the inclusion of the Tkatchenko–Scheffler vdW correction.

Remarkably, the two approaches led essentially to the same conclusion, that is, the inclusion of the vdW correction does not imply substantial changes in the equilibrium interchain length. In the first case, indeed, the calculated values were 3.53 Å and 3.60 Å, respectively with and without the inclusion of Grimme’s vdW correction (see Figure 11), in good agreements with the results of Ref. [52]. In the second case, even though the two chains ended up twisting while remaining parallel, we found similar values (3.46 Å and 3.53 Å) for the equilibrium interchain length, respectively with and without inclusion of Tkatchenko–Scheffler vdW correction. In conclusion, the analysis of the interchain distance alone cannot clarify the role played by the vdW interaction in the case of interest (see [56] for more information). The situation becomes instead clear if we analyse the energy contributions involved in the interchain interaction, as explained in the following.

From an energetic point of view, the strength of the interaction between the helical chains can be estimated by computing the interchain binding energy $E_{b}$, defined by $E_{b}=E_{c}-2E_{s}$, where $E_{c}$ is the total energy of a *couple* of interacting chains and $E_{s}$ that of a *single* isolated chain; in this way, we identify only the energetic contributions related to the interaction between the chains, as it is shown in Figure 11. We can see that, when vdW corrections are included, $E_{b}$ is about 721 meV (the minimum energy of the black solid line in Figure 11), while it reduces to 538 meV when vdW corrections are not considered (the minimum energy of the blue dashed line in Figure 11). Thus, the net difference between these two values (about 183 meV) should be attributed to the vdW correction alone. Here, two interesting points may be underlined: first, the value of $E_{b}$ obtained including the vdW correction corresponds to about 120 meV/atom (we have considered systems with 6 atom per unit cell), somehow compatible with the one calculated for bulk Te-I by Yi et al. [52], with a GGA-PBE functional. Noticeably, it is much larger than the experimentally observed interlayer binding energy of graphite (ranging between 31 and 35 meV/atom [57,58]) or those calculated with different computational approaches and functionals for bilayer graphene (between 18 and 72 meV/atom [59,60,61,62,63,64]). These results underline that Te helical chains are strongly interactive if compared to systems in which a vdW-like force is responsible for the interlayer interaction. Secondly, the vdW contribution to $E_{b}$ appears to be nearly just the 25% of the total, which, on one side, points out that the vdW interaction is clearly not the principal contribution to the interchain interaction; on the other, that the vdW interaction cannot be neglected for this kind of structure.

The relevance of the strongly interacting nature of the chains can be revealed by computing the Electron Localisation Function (ELF), following the analysis given by Koumpouras et al. [65]. The ELF is the probability density of finding another electron near a reference electron with the same spin and it is associated with the electron density of the system. The ELF is a relative (adimensional) measurement of the electron localisation and it takes values between 0 and 1. When it is close to unity (>0.7), the electrons have to be considered as localised (core, covalent bonding regions or lone pairs); on the other hand, when it is in the range between 0.2 and 0.7, the electron localisation is similar to that of the electron gas and hence it is proper of metallic bonds. Here, we compare the ELF of two adjacent Te atoms along the chain with that of the two nearest-neighbours Te atoms between two parallel chains. The ELF is shown in Figure 12 as 2D plots on vertical planes axially cutting a Te-Te bond along a chain or between chains. They manifestly display the different nature of the two bondings, as indicated by the bond lengths found in our calculations: a very localised, covalent-like bonding character along the chains (>0.7) and a weakly localised, metallic-like bonding character between the chains (slightly above 0.2). It is important to stress the fact that, in the latter case, the ELF between the chains is low but not zero, as it would appear for a vdW-like interaction. This residual value of the ELF could be due to the uneven distribution of the highly localised regions of the outer shells, seemingly corresponding to the lone pairs of electrons associated with each Te atom. Indeed, while one pair is localised also along the bond axis (Figure 12b, left atom), the other is moved away from it (Figure 12b, right atom).

Apparently, these results show that the interchain interaction should definitely not possess a covalent (or chemical) character; however, its borderline behaviour (in the terms expressed by Alvarez [54]), together with the arguments brought by Yi et al. [52] and, finally, our results, may lead to think that the lone pairs of the valence electrons allocated between the chains, repelling each other and attracted by these so-created depleted zones, in competition with a vdW-like interaction, could likely cause the interchain distance to reduce, giving life to this hybrid, apparently ambiguous result.

## 4. Conclusions

By means of first-principles calculations, using DFT, we provide novel results in the characterisation of the evolution of the physical properties of three different allotropic forms (α-, β- and γ-phase) of 2D Tellurium (Tellurene), for increasing number of layers.

Our calculations confirm that γ-phase is the most stable in the monolayer configuration, while α-phase is more stable when the number of layer is greater than 1. The calculation of the electronic properties necessarely requires the inclusion of SOC corrections. Overall, the bandgap appears to decrease with increasing number of layer for all the studied phases. Moreover, lower values of the gaps were found, out of symmetry directions, in the case of the α-phase, thanks to a very dense sampling of the Brillouin Zone. The studied systems share a strong optical absorption with characteristic differences between 0 and 4 eV. The 2D exciton model shows that the exciton binding energy tends to decrease by increasing the number of layers. This is due to the related decreasing of the bandgap, accompanied by a more effective electronic screening. Moreover, we generally observe a reduction of the exciton reduced mass for increasing number of layers, which results in an increasing of the exciton radius. Finally, our calculations show that, despite the vdW interaction is not negligible, the equilibrium minimum interchain separation should be mainly attributed to the lone pairs of the valence electrons allocating between the chains.

## Figures and Tables

**Figure 1 nanomaterials-12-02503-f001:**
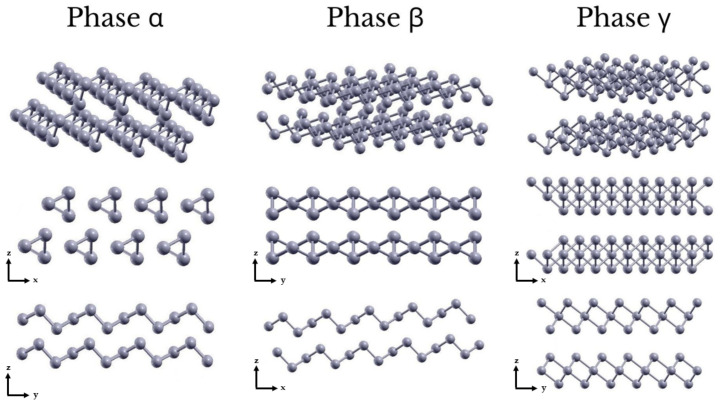
Perspective and side views of the crystal structures of bilayer (2L) α-, β- and γ-phase of Tellurene.

**Figure 2 nanomaterials-12-02503-f002:**
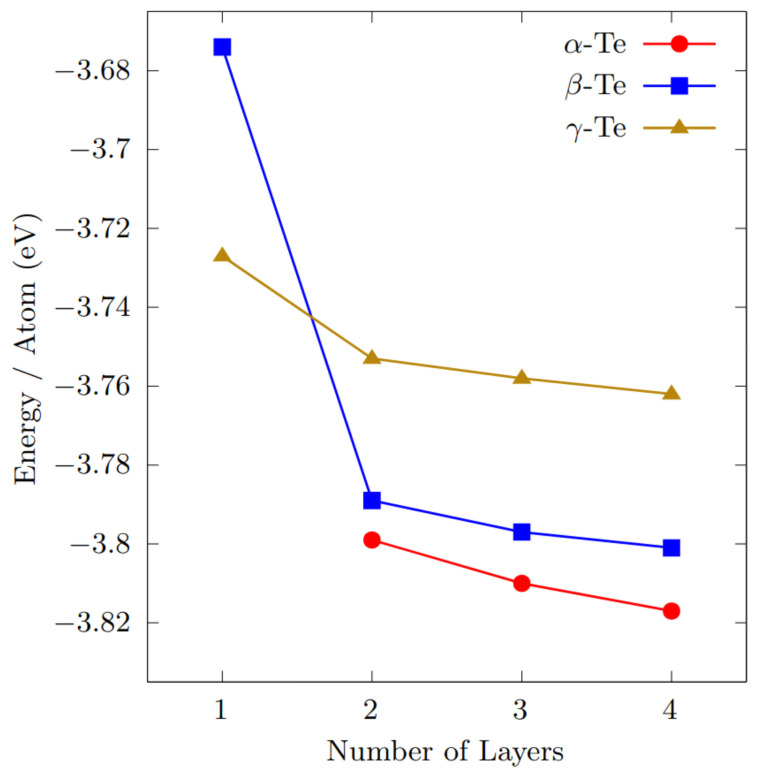
Total energy per atom (rescaled with respect to an isolated Te atom), for increasing number of layers, of the three studied phases. The γ-phase is the most stable in the ML configuration, while the α-phase is preferred for larger layer thicknesses. Note that the ML α-phase is unstable.

**Figure 3 nanomaterials-12-02503-f003:**
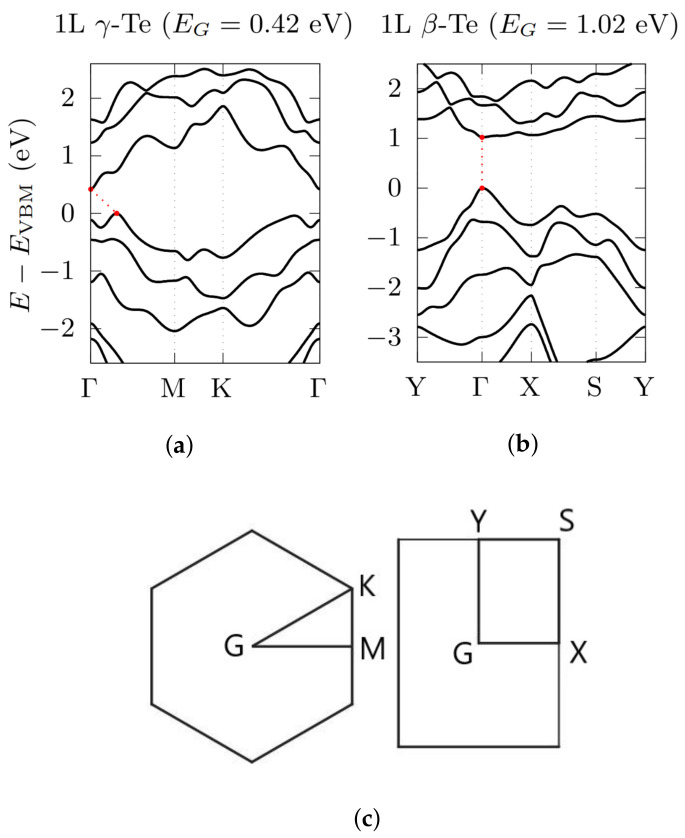
Electronic bandstructures, obtained by using a PBE functional, with the inclusion of SOC, for ML γ-Te (**a**) and β-Te (**b**). Energy rescaled with respect to the VBM. 2D hexagonal (**c**, **left**) and orthorhombic (**c**, **right**) BZ, with the high symmetry points used in electronic bandstructure calculations.

**Figure 4 nanomaterials-12-02503-f004:**
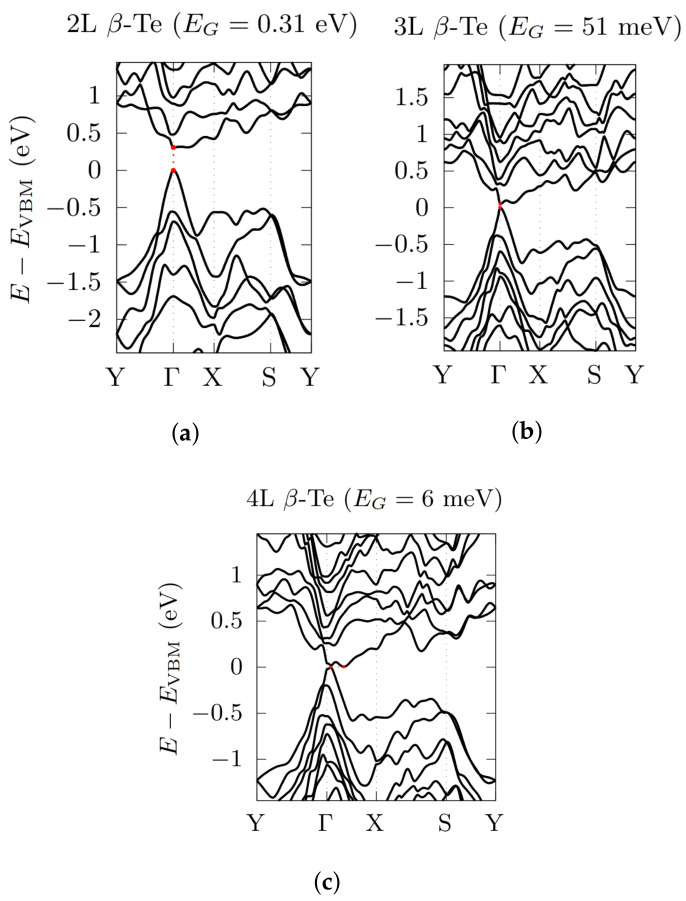
Electronic bandstructures, obtained by using a PBE functional, with the inclusion of SOC, for 2L β-Te (**a**), 3L β-Te (**b**) and 4L β-Te (**c**). Energy rescaled with respect to the VBM. High-symmetry points related to Figure 3 (orthorhombic BZ).

**Figure 5 nanomaterials-12-02503-f005:**
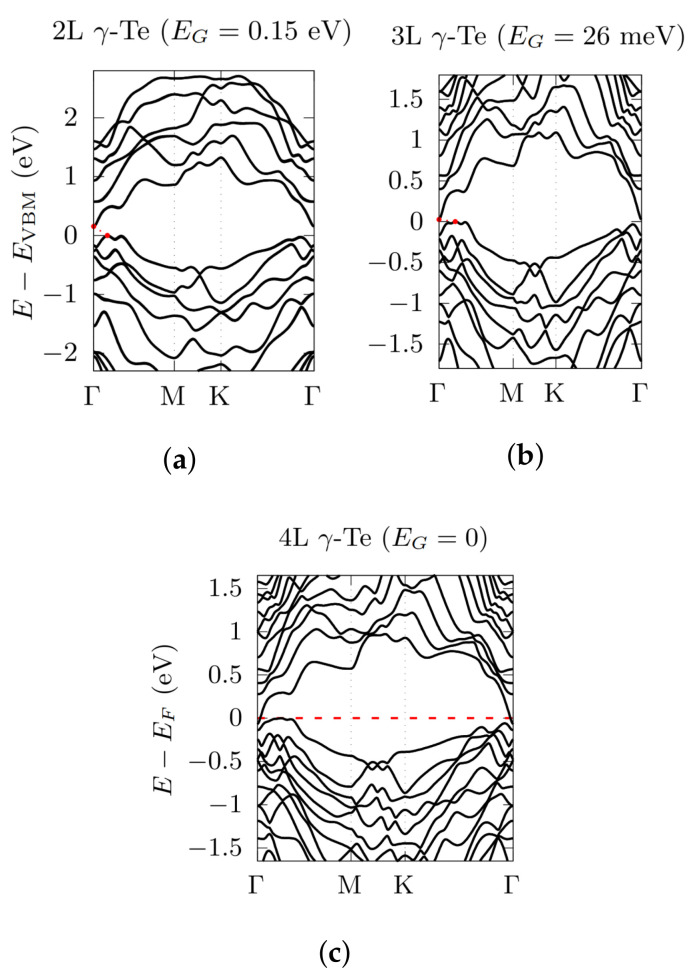
Electronic bandstructures, obtained by using a PBE functional, with the inclusion of SOC, for 2L γ-Te (**a**), 3L γ-Te (**b**) and 4L γ-Te (**c**). Energy rescaled with respect to the VBM (and Fermi energy for 4L). High-symmetry points related to Figure 3 (hexagonal BZ).

**Figure 6 nanomaterials-12-02503-f006:**
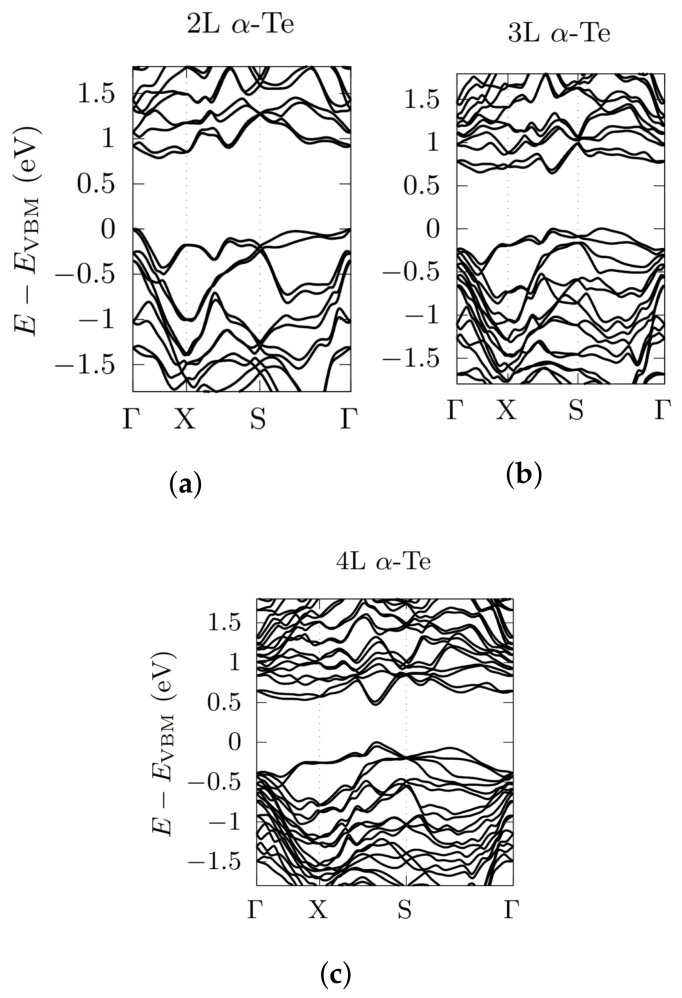
Electronic bandstructures, obtained by using a PBE functional, with the inclusion of SOC, for 2L α-Te (**a**), 3L α-Te (**b**) and 4L α-Te (**c**). Energy rescaled with respect to the VBM. High-symmetry points related to Figure 3 (orthorhombic BZ). Note that lower values of the gaps were found out of high-symmetry directions and they are reported in Table 2.

**Figure 7 nanomaterials-12-02503-f007:**
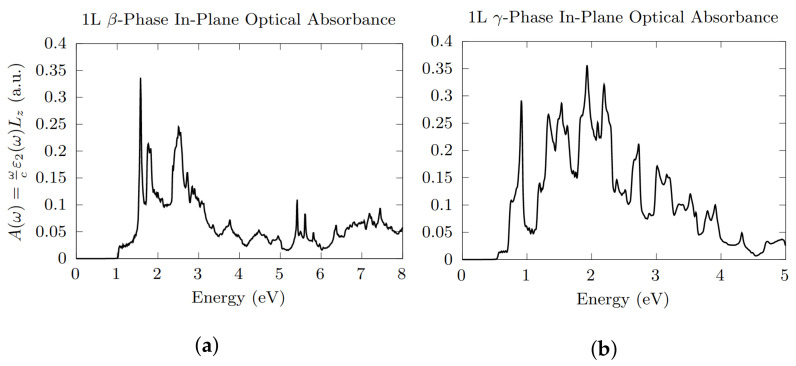
In-plane optical absorbance of (**a**) ML β- and (**b**) γ-Te, with the inclusion of SOC. Absorption energy threshold estimated values of 1.02 eV and 0.54 eV, respectively.

**Figure 8 nanomaterials-12-02503-f008:**
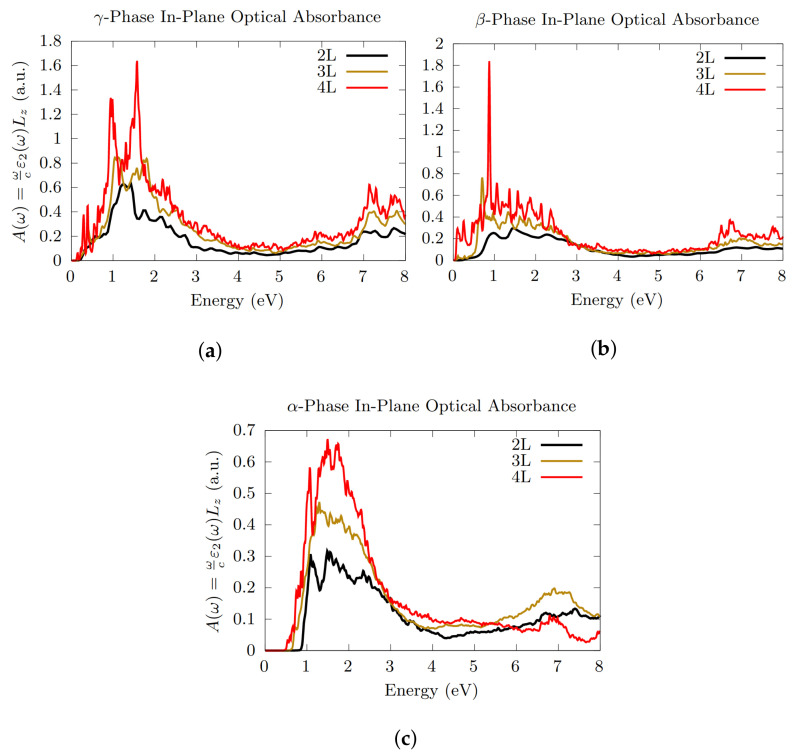
In-plane optical absorbance comparison between 2L, 3L and 4L γ- (**a**), β- (**b**) and α-Te (**c**), with the inclusion of SOC. Overall, the absorption energy threshold decreases for increasing number of layers (see Table 2).

**Figure 9 nanomaterials-12-02503-f009:**
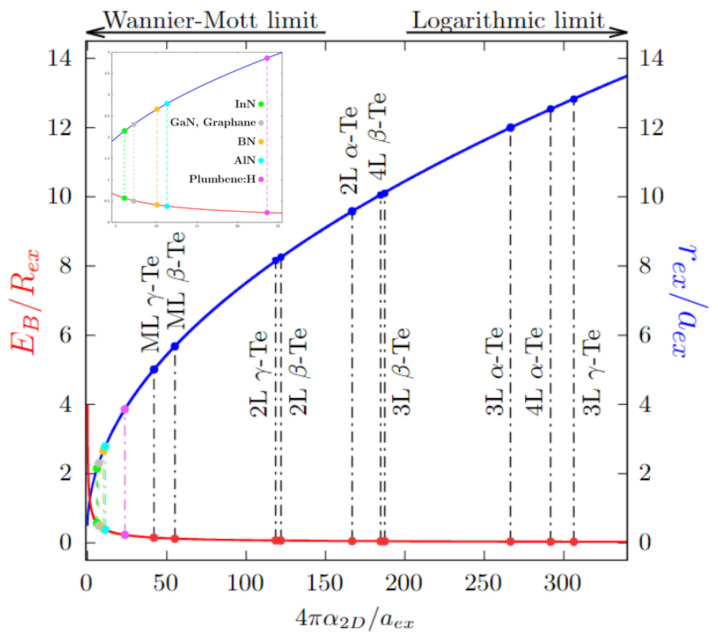
Numerical solutions of the 2D exciton model, for $E_{B}/R_{ex}$ (red) and $r_{ex}/a_{ex}$ (blue), as expressed by Equation (Equation 5), showing the two limits discussed. $E_{B}$ and $r_{ex}$ are the exciton binding energy and radius, respectively; $R_{ex}=R_{H}\frac{\mu }{m}$ is the 3D hydrogenoid Rydberg $R_{H}$, renormalised by the ratio between the effective reduced mass μ and the free electron mass *m*; $a_{ex}=a_{B}\frac{m}{\mu }$ is the Bohr radius $a_{B}$, renormalised by the free electron mass and the effective reduced mass ratio. Results for Te are all from this work. Credits to [37,38,39] for the other results reported (see top left inset): InN (green), GaN and Graphane (grey), BN (orange), AlN (cyan), Plumbene:H (magenta).

**Figure 11 nanomaterials-12-02503-f011:**
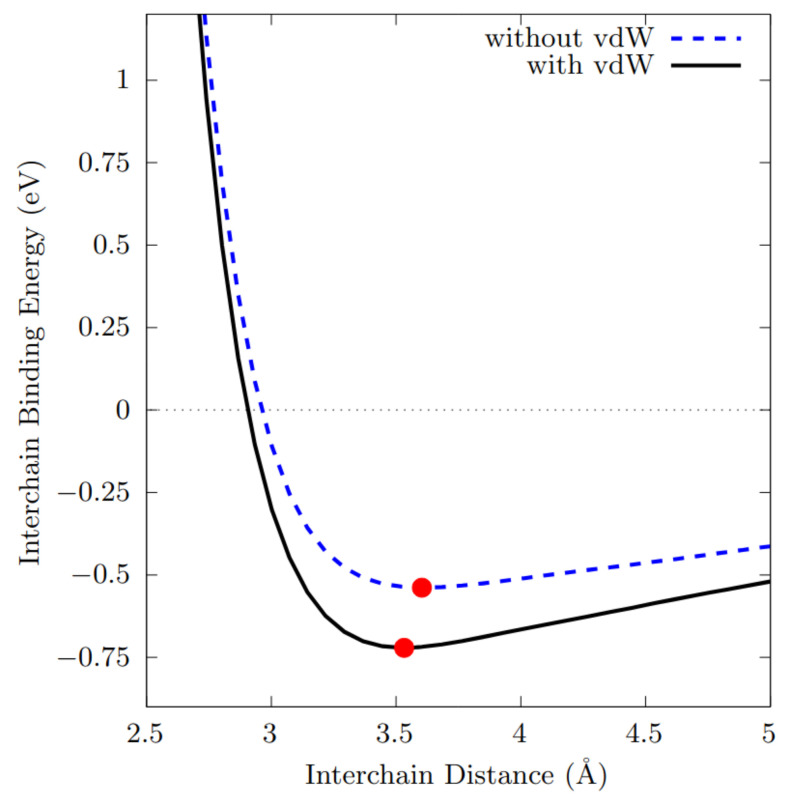
Interchain binding energy as a function of the interchain distance, with and without the inclusion of the Grimme’s DFT-D2 vdW correction. In both case, the chains possess the same fixed geometry. Red dots correspond to the equilibrium interchain separation of minimum energy for the two cases. Energy rescaled with respect to the relative isolated chains (with and without vdW).

**Figure 12 nanomaterials-12-02503-f012:**
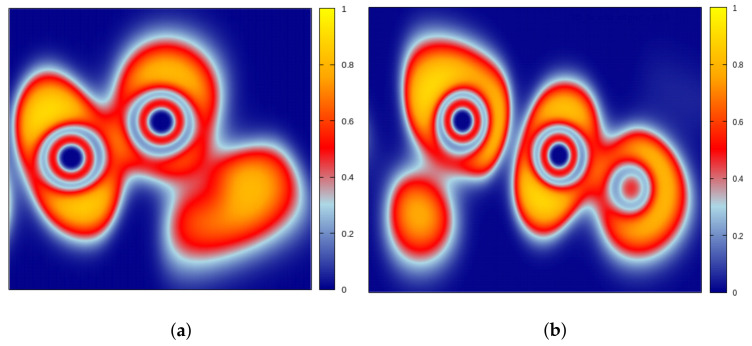
Electron localization function (ELF) for two Te helical chains. Vertical 2D plot cutting through a Te−Te bond axis along a chain (**a**) and between the chains (**b**).

**Table 1 nanomaterials-12-02503-t001:** Optimised lattice parameters (***a*****,**
***b***) and (average) buckling parameter ($d_{z}$), for increasing number of layers, of the three studied phases. Results from the literature are also reported below each related value. Note that ML α-Te is unstable. In the case of γ-phase, a=b.

	1L		2L		3L		4L	
	*a*, *b* (Å)	$d_{z}$ (Å)	*a*, *b* (Å)	$d_{z}$ (Å)	*a*, *b* (Å)	$d_{z}$ (Å)	*a*, *b* (Å)	$d_{z}$ (Å)
α-phase								
This work	-	-	5.79, 4.23	2.10	5.88, 4.28	2.08	5.92, 4.30	2.11
[30]			5.80, 4.27		5.88, 4.31		5.92, 4.34	
β-phase								
This work	5.48, 4.17	2.17	5.81, 4.18	2.08	5.92, 4.20	2.04	5.96, 4.20	2.03
[14,45]	5.49, 4.17	2.16	5.71, 4.13		5.81, 4.13		5.85, 4.14	
γ-phase								
This work	4.15	3.68	4.19	3.71	4.19	3.71	4.20	3.72
[14]	4.15	3.67						

**Table 2 nanomaterials-12-02503-t002:** Calculated DFT electronic bandgaps, for increasing number of layers and with the inclusion of SOC, of the three studied phases. When the bandgap is indirect, we also report the direct bandgap value (in square brackets). Note that ML α-Te is unstable and 4L γ-Te is metallic.

	1L	2L	3L	4L
	$E_{G}$ (eV)	$E_{G}$ (eV)	$E_{G}$ (eV)	$E_{G}$ (eV)
α-phase	−	0.67 [0.86]	0.50 [0.62]	0.42 [0.48]
β-phase	1.02	0.31	0.051	0.006 [0.075]
γ-phase	0.42 [0.54]	0.15 [0.26]	0.026 [0.19]	0

**Table 3 nanomaterials-12-02503-t003:** Exciton binding energy ($E_{B}$) and radius ($r_{ex}$), as calculated within the Rytova-Keldysh model, compared to reported GW-BSE results in literature ($E_{B}^{BSE}$). Extrapolated exciton reduced mass renormalised with respect to the free electron mass (μ/m) is also reported.

	$\frac{\mu }{m}$	$E_{B}$ (eV)	$r_{ex}\;\left(Å\right)$	$E_{B}^{\mathrm{BSE}}$ (eV)
2L α-Te	0.36	0.26	15	
3L α-Te	0.34	0.17	19	
4L α-Te	0.26	0.12	25	
1L β-Te	0.34	0.57	9	0.57 [46]
2L β-Te	0.17	0.15	26	
3L β-Te	0.09	0.06	57	
4L β-Te	0.02	0.01	267	
1L γ-Te	0.10	0.20	27	0.15 [50]
2L γ-Te	0.07	0.07	63	0.10 [50]
3L γ-Te	0.13	0.06	52	0.07 [50]

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
