# Peer review of "Evolution of the Electronic and Optical Properties of Meta-Stable Allotropic Forms of 2D Tellurium for Increasing Number of Layers"

_nanomaterials, 2022, doi:10.3390/nano12142503_

Round 1
Reviewer 1 Report
The authors report a systematic computational study of Tellurine stackings for three different allotropic forms. The authors carefully select the three, forms based on their relevance, which they clearly state, and proceed to study the evolution of the electronic structure and optical properties as a function of the number of layers. The study is very detailed, and the authors make a fantastic job in placing their work within the work of others providing a clear picture for these systems. I think it is an excellent paper and I recommend publication once the comments below are addressed. The paper is scientifically sound, rigorous, and relevant to the field, therefore, although many and wordy, these are minor comments.
1) The authors compare the lattice parameters they obtained in their calculations with the literature. Consider adding those published values to table 1 (maybe in parenthesis by the corresponding value reported here).
2) Page 4, first line in 3.2 I think “…have been calculated for an increasing number of layers…” sounds better.
3) In page 4, the authors indicate that they only report simulations with SOC “for the sake of simplicity” clearly that is not the reason as the results are clearly affected by this choice, as they highlight when describing their results and it is not until the conclusions that they recognize the need for SOC. They should use the significant differences the describe between calculations with and without SOC to note, in the body of the paper, not just in the conclusions, that SOC is necessary
4) Page 9, first line in section 3.3. Consider rephrasing “We now move on to discuss…” or simply “Now we discuss…”
5) The same vertical scale should be used in both plots in figure 7 to help with the comparison. Even better, the two optical spectra can be plot in the same plot.
6) In page 10, line 195, the reference to figure 9 is incorrect, it should be to figure 7 (actually, the authors can be more precise and refer to figure 7a).
7) In page 10, the paragraph below figure 7, the authors can also be more precise and refer to each individual sub figure, figure 8a) and figure 8b) respectively for the two times this figure is cited in that page. In page 11, the citation should be to figure 8c.
8) The authors explain that the reduction of the energy gap with the number of layers leads to a lowering of the absorption threshold for -Te and -Te, agree and it is clearly observed in figures 8b) and 8c), so that is fine. However, the gap for -Te also decreases, but the authors do not mention anything about it, and it is hard to see, looking at figure 8a), if that is happening. So does the absorption threshold decreases for -Te as well? If so (I suspect it does), they should say so, so if not, why not?
9) I suggest the authors used dashed lines for one of the curves in figure 10 or otherwise make them clearly different so they can be told apart in grayscale. Of course, VdW is an attractive force, therefore the energy will be lower when VdW is included, but still, it won’t hurt to make sure the reader can tell which curve is which.
10) Certainly, the authors can choose the way to tell their story, however, the role of the VdW interaction in the layer-layer binding is clear just looking at figure 10, however, the authors choose to take the long way. They focus on the equilibrium distance from figure 10 and conclude that it does not provide the answer, then they analyze the nature of the bond and conclude that this study does not either, only then, they go back to figure 10 and discuss the obvious different in the interaction energy when VdW is not considered vs when it is considered. Again, the authors can report their story in any way they want and indeed discussing the layer-layer distance and the nature of the bond is interesting in their own right, but I would suggest to first discuss the energetic contribution from VdW and only then expand to those other aspects.
Author Response
Dear referee,
Thank you for your kind review. Please see the attachment.

Reviewer 2 Report
2D tellurium is an emerging nanomaterial with potential wide applications for (opto-)electronics devices. The authors report timely here the electronic and optical properties of 2D tellurium thin film with different allotropes (i.e., phases) and lay numbers using the density functional theory calculations. Such calculations present a detailed understanding of band gap engineering by thickness, exciton binding energy and radius, and layer interaction. The current manuscript can be further improved by addressing the following comments:
First, both the electronic and optical properties are calculated using the PBE method, which underestimate the size of band gap. Please comment how this would affect the optical absorption spectrum compared to the more advanced GW-BSE approach.
Second, how would the DOE is related to the optical absorption spectra?
Third, it is claimed that ML b-Te shows strong anisotropic absorption, but there is no data to support this statement.
Last, in line 195, please check if this is a typo and it should be Fig.7a.
Author Response

(The authors gave the same response as above.)

Reviewer 3 Report
The manuscript presents the results of a skillful study on the 2D Tellurine a-, b-, and g-phases. In particular, the dependence of their electronic and optical properties on the number of layers.
The text consists of three strata.
The first, presents the results of a somewhat standard set of periodic DFT calculations, including band-structures and optical properties. Additional phonon calculations might have revealed the meta-stability of the respective structures.
The second, compares the DFT calculations to the results from a Keldysh-Rytova exciton model and gives interesting insights into the exciton binding energies.
The third, pinpoints interesting questions and gives partial answers related to the nature of the inter-layer interaction, its relation to the van-der-Waals interaction, and electronic lone pairs.
Some minor further editing of the text would be helpful. In particular, the title might be slightly revised as the word “evolution” has strange connotations in the current context. Probably, “Dependence of … on the Number of Layers” would sound better.
Author Response

(The authors gave the same response as above.)
